# Addressing the barriers to peritoneal dialysis— Visual appeal matters

**Sam Stephens**[1], **Larissa Whale**[1], **Makenzie Kapales**[1], **Fatima Ayub**[2], **Morten O. Jensen**[2,3], **Manisha Singh** [iD][2]*

**1** Department of Biomedical Engineering, University of Arkansas, Fayetteville, Arkansas, United States of America, **2** Division of Nephrology, Department of Internal Medicine, University of Arkansas for Medical Sciences, Little Rock, Arkansas, United States of America, **3** Department of Surgery, University of Arkansas for Medical Sciences, Little Rock, Arkansas, United States of America

* msingh@uams.edu

## Abstract

Peritoneal dialysis (PD) is an effective renal replacement strategy for patients with end-stage renal disease utilizing the peritoneum as the filter and PD catheter as access. A survey of PD patients showed that some felt uncomfortable with the length of current catheters and would be interested to explore newer, shorter designs. We redesigned the transfer set and external portion of the catheter, addressing this barrier as a part of our multi-institutional design project led by a nephrologist from the University of Arkansas for Medical Sciences (UAMS) and an engineering design team from the biomedical engineering department at the University of Arkansas. Multiple designs were considered, including spiral, retractable, and collapsible bulbs, with an accordion-style mechanism being selected for prototyping. We created several prototypes, first by 3D printing as well as by silicone casting. Computational fluid analysis showed the design to be fully capable of delivering clinically relevant flows. The final design of our PD transfer set has a flexible accordion section that is 7 cm when extended and collapses to substantially shorten this length. We propose that the design can also be extended to the extra-abdominal section of the PD catheter.

## Introduction

Peritoneal dialysis (PD) is a method of renal replacement therapy that utilizes the patient's peritoneal lining as the filter with a PD catheter to access this filter [1]. The PD catheter additionally has a transfer set attached for protection during routine manipulations of dialysis [2]. PD offers a patient multiple benefits including patient autonomy and potentially increased survivability [3–5]. Despite this, multiple barriers exist to its use. We identified one of the barriers (patient-related factor) to be the unnecessarily long PD transfer set protruding from their abdomen and affecting their self-image and daily life [6–8]. Between the catheter and transfer set, many PD patients have upwards of 16 cm of tubing protruding from their abdomen. Often,

**Data availability statement:** All relevant data are either within the manuscript or are publicly available at: https://github.com/SamStephensUARK/Peritoneal-Dialysis-Catheter.

**Funding:** The author(s) received no specific funding for this work.

**Competing interests:** The authors have declared that no competing interests exist.

young patients elect to forgo PD due to this exposed tube length. A patient-centered survey of PD patients assessing the need for a modified PD tubing set was conducted, revealing that most patients wanted to try one that could be smaller, prompting the need to re-design the PD catheter transfer set tube.

A retractable catheter and transfer set, whose lengths may be shortened when not in use, was designed to address these concerns to improve the PD use. Our design project aimed to provide an appealing and relatively inconspicuous design for PD patients without compromising the flow, starting with the transfer set portion of the tube. We aimed to design a new transfer set that can retract to 3–5 cm when not in use. The modified transfer set retains the functionality of the current PD transfer set design.

## Methods

### Design and modeling

Several mechanisms were considered for catheter and transfer set expansion/retraction. The transfer set and externalized portion of the catheter were treated identically, as these are both conventionally composed of regular, straight tubing. Only the external portion of the PD catheter is changed, with the portion extending into the peritoneal cavity being unaltered. The designs considered are hereafter referred to as accordion, flat spiral, helix, and telescoping. Three-dimensional computer-aided design (CAD) models were created for several different catheters and transfer set versions. All models were generated using SolidWorks (Dassault Systemes, Vélizy-Villacoublay, France). Inner and outer diameters of three and five millimeters respectively were used for all designs, except for the telescoping versions whose diameters vary along the length. The various expansion mechanisms are illustrated in Fig 1.

The accordion design is reminiscent of the geometry found within the flexible portion of straws that can bend to establish an angle between the two openings. Expansion/retraction is achieved by applying tensile or compressive force axially along the tube's length axis. The wave-like structures are designed such that the expanded/retracted tube will "lock" into place, remaining stable at either length [9].

The flat spiral is designed such that the retracted tube coils upon itself within the same plane. Expansion is accomplished by applying a tensile force upon the tube's distal end, causing the coil to unfurl. While expanded, the coil will not necessarily remain along a plane but rather extend into the direction normal to the coiled plane. This design has the advantage that the coiled tube's plane may be placed along the patient's abdomen, presenting a minimally visible disturbance to their clothing and appearance.

The helix design places the external portion of the catheter/transfer set in a helical configuration about a straight axis. The tubing is manufactured such that the length is minimized/retracted in the normal state, with no applied stress. Expansion is achieved by applying a tensile force along the helix's axis. The degree to which the length is increased per unit force applied is determined by the helix diameter and pitch, or the linear distance between adjoining coils.

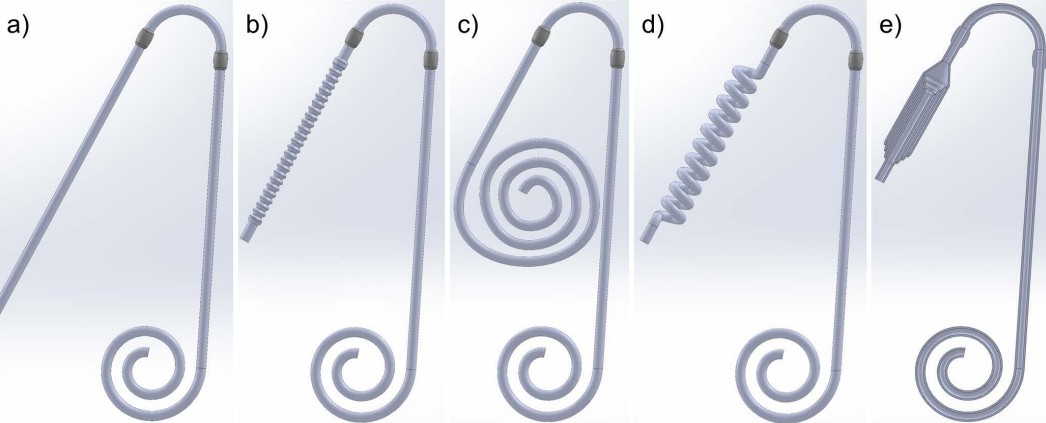

**Fig 1. Various designs for extendable catheters, including: a) conventional (non-extending), b) "accordion," c) flat spiral, d) helix, and e) telescoping (retracted).**

The telescoping design utilizes a set of either rigid or semi-rigid segments capable of telescoping within one another. The segments are formed such that when fully extended, they create a leak-tight seal.

While all expansion mechanisms have potential to achieve the goal of minimizing tube length when not in use, the accordion design was selected as the most feasible for consistent, scalable fabrication.

## Physical prototype

The first iteration of prototype production utilized a 3D printed model of the accordion transfer set using a Formlabs Form 2 resin printer (Formlabs, Somerville, MA) using the Elastic 50A resin. Formlabs Elastic resin produces flexible, semi-transparent parts with a stiffness comparable to silicone. Following printing, the transfer set was washed with iso-propyl alcohol and cured under UV light in the Form Cure. The prototype featured a design composed of a central accordion mechanism, two cylindrical ends, and two tapered connectors. The cylinders on each end of the collapsible section encourage collapsibility and stability, while the connectors allow the transfer set to connect to the catheter on one end and the dialysis machinery on the other.

Given issues with scaling 3D printing to manufacturing-level production, and to ensure our prototypes more closely align with conventionally manufactured catheters, a second prototype was created through a molding process [10,11]. The mold, illustrated in Fig 2, consisted of three parts – two identical external mold halves (one shown in Fig 2a) and one core (Fig 2b)- that were each 3D printed on a Form 2 resin printer using Formlabs Gray resin. This resin produces rigid parts with excellent geometric accuracy. The core represents the negative space inside the transfer set and was designed to include a hollow interior and extrusion mechanism at its end into which the silicone could be injected into the assembled mold. Small cylindrical pegs at each end of the core index into recesses in the external molds to ensure the core remains centered within the cavity. The geometry produced by the mold is shown in Fig 2c, as illustrated by the section view of the transfer set. Casting was performed using a 2-part molding silicone introduced into the mold through the core's central channel.

## Computational fluid dynamics

To evaluate flow through the expandable catheter design, a computational fluid dynamics (CFD) simulation was performed using SolidWorks Flow Simulation 2022. The CAD model for the accordion design was used as the geometry, with the entire interior volume taken as the fluid domain. Adiabatic, rigid walls were used with 5-micron surface

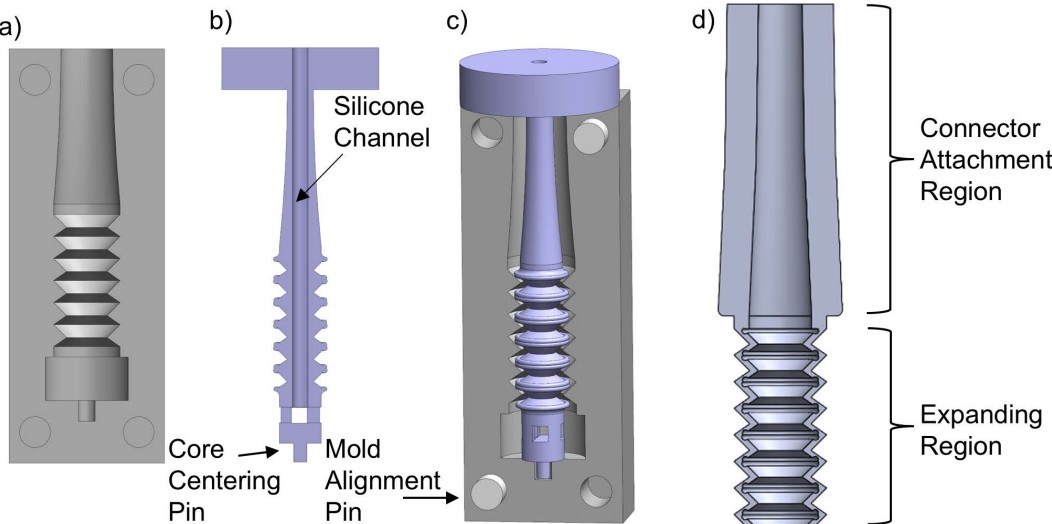

**Fig 2. Transfer set molding system, showing a) external mold, b) core (section-view) with silicone channel and extrusion mechanism, and c) resulting detail of the accordion mechanism and transition to connector region.** For illustrative clarity, only a small segment of the transfer set's expandable region is shown.

roughness. The fluid domain contains a single inlet and multiple outlets comprised of 23 small openings evenly spaced around the peritoneal cavity segment. To simplify the computational domain, catheter/transfer set connectors were not considered in the CFD analysis. Based on literature values for intraperitoneal cavity pressure, outlet pressure of 1.57 kPa (16 cmH$_2$O) was prescribed [12,13]. Inlet pressure was calculated from the hydrostatic pressure exerted by a 1 m column of fluid, representing the pressure exerted by a dialysate bag suspended from an elevated pole.

$$P = \rho g h$$

The hydrostatic pressure P due to a fluid column is given above, where $\rho$ is the density of the working fluid (1000 kg/m3), g is the acceleration due to gravity (9.81 m/s2), and h is the height of the fluid column (1 m). A pressure of 9.8 kPa (100 cmH$_2$O) was used for the CFD inlet pressure. Inlet pressure was held constant as the changes in hydrostatic pressure from decreasing fluid level within the bag are small relative to overall pressure, and furthermore vary with bag dimensions.

Simulations were evaluated at the steady state with zero initial fluid velocity. It is assumed that the pressure from the hydrostatic fluid column remains the same as the fluid velocity increases.

$$Re = uD/\nu$$

Reynolds number, denoted as Re, describes how turbulent a given flow is expected to behave. It is a function of fluid velocity u, hydraulic diameter D and kinematic viscosity $\nu$. Fluid velocity is calculated from the volumetric flowrate and the flow's cross-sectional area. Considering a typical real-world flow rate of approximately 0.2 L/min, the Reynolds number of such a flow through a 3 mm diameter cavity with a kinematic viscosity approximately equal to that of water at room temperature would be approximately 1,400. As this Reynolds number falls within the laminar regime, simulated flows were assumed to be laminar, and turbulent flow was not considered.

## Results

### Physical prototype

The 3D printed molding system (with one external mold removed for clarity) is shown in Fig 3a. An example segment of the accordion transfer set, cast using the molding system, is presented. This segment has been cut to reveal a section view, demonstrating the thin, consistently cast geometry. While creating consistent, full-length transfer set prototypes proved difficult and would likely require industrial-level control and tuning, smaller-scale production was successful.

The resin-printed prototype was successful, illustrated in Fig 3c. The design of the collapsible, "bendy" straw portion featured two cones: a larger top cone at an angle of 45° from the center and a smaller bottom cone at 35° from the center. The internal radii were 1.75 mm and 3.00 mm, with respective external radii of 2.46 mm and 3.71 mm. At the apex of the largest diameter, both top and bottom cones had a small, semi-circular indentation cut to promote collapsibility; the semi-circle radius was half of the smallest wall thickness. The total length of the extended accordion portion was 70 mm, comparable to the exposed tubing of current transfer sets on the market.

### Computational fluid dynamics

The CFD simulation, which utilized realistic boundary conditions, yielded favorable pressures and velocities within the catheter's lumen. Integration of the outlet flux across all peritoneal segment openings yielded a total flow rate of 0.57 L/min. Streamlines and fluid pressure for the simulated flow may be seen in Fig 4. The simulations show that the flow through this catheter is comparable to the current models.

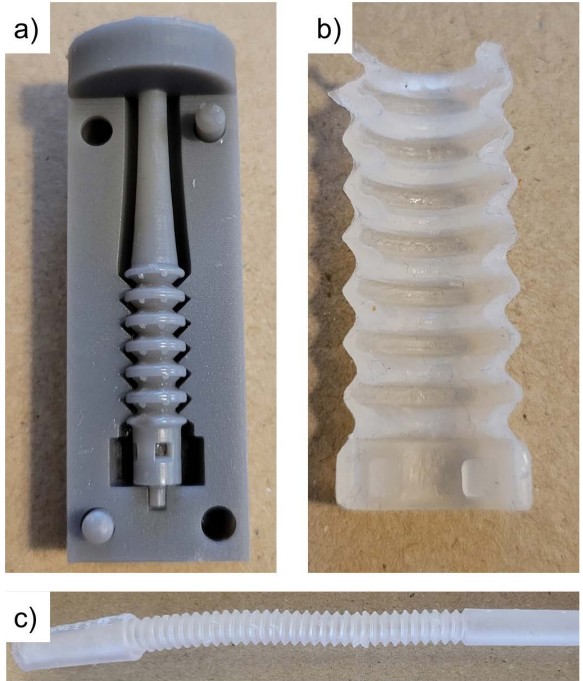

**Fig 3. Physical prototypes, illustrating a) 3D printed molding system, b) cast transfer set segment, shown in section-view to illustrate geometry, and c) 3D printed transfer set.**

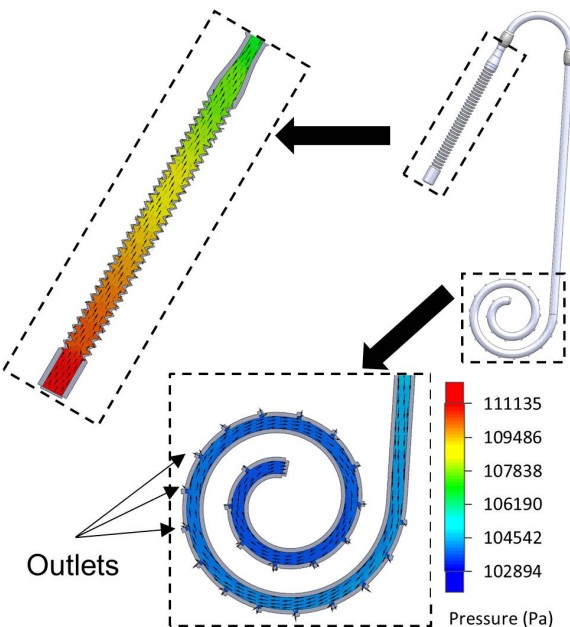

**Fig 4. Flow simulation results for accordion catheter model.** Fluid pressure is shown by color gradient with fluid velocity indicated by yellow vectors. The model is shown in a partial section view.

## Discussion

The global peritoneal dialysis market size stood at USD 3,589.9 million in 2017 and is projected to reach USD 6,077.2 million by 2025 [14]. The PD market is expected to grow by 6.9% annually due to the rise of end-stage renal disease (ESRD) patients, the surge in demand for at-home dialysis treatment, and technological advances in peritoneal dialysis (Peritoneal dialysis market size, share). This growth, coupled with patients' resistance to PD's aesthetically unattractive catheters, illustrates the need for improved designs. To the best of our knowledge, there is no alternative PD transfer set with a reduction of length on the market, making these designs a novelty within the clinical space. Transfer sets are usually 16–18 cm long and can be uncomfortable for patients. In addition, the transfer set can also affect patients' internal issues, such as reduced self-esteem. While the catheter cannot be entirely eliminated, and there would still be an exposed tube coming out of the belly, it is, however, substantially shorter. Our patients do not have the choice of having a catheter or not, but a choice of a shorter catheter may go a long way in addressing a barrier and increasing quality of life.

The final prototype successfully fulfilled the goal of designing a relatively inconspicuous, more aesthetic PD catheter, starting with a transfer set that retracts to a shorter length of around 3–5 cm while retaining the function of a typical PD transfer set. This retractable design can be applied to the external portion of the catheter to reduce the length further.

## Limitations

While physical prototyping was successful from a proof-of-concept standpoint, the examples produced were not clinical grade. High-grade silicone as well as industrial production controls would be necessary to translate to a clinic-ready product.

Risk for exit site infection, the most significant concern for catheters, was not considered. The retractable design is not anticipated to impact the risk of patients developing an exit site infection as similar materials and implantation techniques would be employed to clinical catheters. Further bacteriological study should be conducted prior to clinical trials however.

Only the infusion stage of treatment is simulated in this work. However, given that the simulated flowrates are well within the ranges of those of real-world catheters, it is reasonable to expect that the drainage flows/times would be as well.

## Acknowledgments

The authors thank Jose Sanchez Gracias, Christopher Schaefer, Andrew Tegel, and Tayte Stephens for assistance with physical prototyping. Additionally, we are grateful to the University of Arkansas Biomedical Engineering Department for lab assistance and DCI (Dialysis Clinic, Inc) for conducting the patient survey. The design is protected by a pending patent through BioVentures, LLC, and approved by the US. Patent and Trademark Office (63/404,799)

## Author contributions

**Conceptualization:** Larissa Whale, Makenzie Kapales, Fatima Ayub, Morten O. Jensen, Manisha Singh.

**Formal analysis:** Sam Stephens, Larissa Whale, Makenzie Kapales.

**Investigation:** Sam Stephens, Larissa Whale, Makenzie Kapales, Manisha Singh.

**Methodology:** Sam Stephens, Larissa Whale, Makenzie Kapales.

**Project administration:** Morten O. Jensen.

**Resources:** Morten O. Jensen, Manisha Singh.

**Supervision:** Fatima Ayub, Morten O. Jensen, Manisha Singh.

**Validation:** Sam Stephens, Larissa Whale, Makenzie Kapales, Manisha Singh.

**Visualization:** Sam Stephens, Larissa Whale, Makenzie Kapales.

**Writing – original draft:** Sam Stephens, Larissa Whale, Makenzie Kapales, Fatima Ayub, Manisha Singh.

**Writing – review & editing:** Sam Stephens, Larissa Whale, Makenzie Kapales, Fatima Ayub, Morten O. Jensen, Manisha Singh.

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
