## [Decision Letter · Decision Letter 0]

Dear Dr. Singh,

Thank you for submitting your manuscript to PLOS ONE. After careful consideration, we feel that it has merit but does not fully meet PLOS ONE’s publication criteria as it currently stands. Therefore, we invite you to submit a revised version of the manuscript that addresses the points raised during the review process.

We look forward to receiving your revised manuscript.

Kind regards,

Ankur Shah

Academic Editor

PLOS ONE

Journal Requirements:

3. We note that your Data Availability Statement is currently as follows: [All relevant data are within the manuscript and its Supporting Information files.] Please confirm at this time whether or not your submission contains all raw data required to replicate the results of your study. Authors must share the “minimal data set” for their submission. PLOS defines the minimal data set to consist of the data required to replicate all study findings reported in the article, as well as related metadata and methods (https://journals.plos.org/plosone/s/data-availability#loc-minimal-data-set-definition). For example, authors should submit the following data: - The values behind the means, standard deviations and other measures reported; - The values used to build graphs; - The points extracted from images for analysis. Authors do not need to submit their entire data set if only a portion of the data was used in the reported study. If your submission does not contain these data, please either upload them as Supporting Information files or deposit them to a stable, public repository and provide us with the relevant URLs, DOIs, or accession numbers. For a list of recommended repositories, please see https://journals.plos.org/plosone/s/recommended-repositories. If there are ethical or legal restrictions on sharing a de-identified data set, please explain them in detail (e.g., data contain potentially sensitive information, data are owned by a third-party organization, etc.) and who has imposed them (e.g., an ethics committee). Please also provide contact information for a data access committee, ethics committee, or other institutional body to which data requests may be sent. If data are owned by a third party, please indicate how others may request data access.

Additional Editor Comments:

This is an interesting manuscript and addresses an important questions. There are some very important comments from the reviewers that need to be addressed and will improve the manuscript further. Please address all these comments when submitting the revised version of the manuscript.

Reviewers' comments:

Reviewer's Responses to Questions

**Comments to the Author**

1. Is the manuscript technically sound, and do the data support the conclusions?

Reviewer #1: Yes

Reviewer #2: Partly

2. Has the statistical analysis been performed appropriately and rigorously?

Reviewer #1: Yes

Reviewer #2: No

3. Have the authors made all data underlying the findings in their manuscript fully available?

Reviewer #1: Yes

Reviewer #2: Yes

4. Is the manuscript presented in an intelligible fashion and written in standard English?

Reviewer #1: Yes

Reviewer #2: Yes

Reviewer #1: At a time when prescribing treatment must take patient perception into account, it is important to consider modifications in peritoneal dialysis transfer sets that can minimize the impact on body image.

The study is well conducted and the proposed solutions are interesting, but it is an in vitro simulation that only considers half of an exchange. In fact, the drainage phase preceding the infusion phase takes place under different hemodynamic conditions: the height between the abdomen and the drainage bag is not two meters, but usually a maximum of 1m. The authors would therefore need to simulate the drainage phase of the peritoneal cavity. In addition, the intraperitoneal pressure of 16 cm is that of the peritoneal cavity filled with 2 liters, but it decreases during the exchange. The design of the extension line, in the drainage phase, will not be the same as when it is fully extended and connected to a bag at a height of 2 m. For peritoneal dialysis patients, the amount of spent time during bag exchanges is almost always related to drainage time, not infusion time. The authors' work should be complemented by a drainage simulation, taking into account the evolution of the line shape in the declive position.

A subsequent study, prior to any human trials, should take into account the bacteriological aspect to ensure that the change of design does not modify the “flush before fill” effect.

Reviewer #2: "Long PD transfer set protruding from their abdomen and affecting their self-image and daily life" - the cited studies don't mention catheter length as being a significant concern for PD patients. Reference 6 stated < 5% cited the presence of a catheter as being a factor against the decision to pursue PD which does not seem to support "many."

As the height of the fluid column will change as dialysate flows from the dialysate bag to the patient's abdomen, should the calculations for hydrostatic pressure be reworked to reflect a dynamic fluid column rather than a static one?

Clarification needed for "typical real-world flow rate of approximately 0.2 L/min (lines 157-158); is this due to the 3mm cavity?

The paper notes that the CFD was only on the helical prototype based on the CAD model for the helical design being used, but the conclusion as written appears to advocate for the accordion/expandable prototype, which was not subjected to a CFD.

**Do you want your identity to be public for this peer review?** For information about this choice, including consent withdrawal, please see our Privacy Policy

Reviewer #1: No

Reviewer #2: No

---

## [Author Response · Author response to Decision Letter 1]

9 May 2025

Dear Plos One editorial team,

Thank you for taking the time to examine our manuscript and provide constructive feedback.

We value the input and have adjusted the manuscript accordingly. Primarily, we have added clarity to elements of the Methods, clarifying the simulation section especially. We expand upon previous description and also add discussion to justify assumptions made in conducting the simulations. Additionally, we clarify and qualify several statements within the Introduction and Limitations.

Below are responses to the reviewer’s comments in bold font, new additions are highlighted yellow.

Reviewer #1:

• At a time when prescribing treatment must take patient perception into account, it is important to consider modifications in peritoneal dialysis transfer sets that can minimize the impact on body image.

The study is well conducted and the proposed solutions are interesting, but it is an in vitro simulation that only considers half of an exchange. In fact, the drainage phase preceding the infusion phase takes place under different hemodynamic conditions: the height between the abdomen and the drainage bag is not two meters, but usually a maximum of 1m.

The reviewer is absolutely correct in pointing this out, and the simulations and text have been amended to utilize a hydrostatic pressure from a bag elevated 1m above the abdomen.

“Inlet pressure was calculated from the hydrostatic pressure exerted by a 12 m column of fluid, representing the pressure exerted by a dialysate bag suspended from an elevated pole.”

“The hydrostatic pressure P due to a fluid column is given above, where ρ is the density of the working fluid (1000 kg/m3), g is the acceleration due to gravity (9.81 m/s2), and h is the height of the fluid column (12 m). A pressure of 9.8 kPa (100 cmH2O) was used for the CFD inlet pressure.”

• The authors would therefore need to simulate the drainage phase of the peritoneal cavity. In addition, the intraperitoneal pressure of 16 cm is that of the peritoneal cavity filled with 2 liters, but it decreases during the exchange. The design of the extension line, in the drainage phase, will not be the same as when it is fully extended and connected to a bag at a height of 2 m. For peritoneal dialysis patients, the amount of spent time during bag exchanges is almost always related to drainage time, not infusion time. The authors' work should be complemented by a drainage simulation, taking into account the evolution of the line shape in the declive position.

Thank you for this insightful comment, the reviewer is correct that only the infusion flow is simulated. To accurately model drainage, the functional relationship of pressure with time must be known to prescribe “inlet” and “outlet” pressures (here, inlet and outlet would be reversed from the simulation we show). Modeling such a time-varying pressure is additionally beyond the scope of the work conducted here. However, as the infusion simulation produces a volumetric flow rate consistent with real-world infusion flows/times, it is reasonable to expect similarly appropriate drainage times. This point has been added to the Limitations section as follows:

“Only the infusion stage of treatment is simulated in this work. However, given that the simulated flowrates are well within the ranges of those of real-world catheters, it is reasonable to expect that the drainage flows/times would be as well.”

• A subsequent study, prior to any human trials, should take into account the bacteriological aspect to ensure that the change of design does not modify the “flush before fill” effect.

The reviewer raises an important point here and is definitely correct. We do address this briefly in the Limitations section, but have expanded this for emphasis.

“Risk for exit site infection, the most significant concern for catheters, was not considered. The retractable design is not anticipated to impact the risk of patients developing an exit site infection as similar materials and implantation techniques would be employed to clinical catheters. Further bacteriological study should be conducted prior to clinical trials however.”

Reviewer #2:

• "Long PD transfer set protruding from their abdomen and affecting their self-image and daily life" - the cited studies don't mention catheter length as being a significant concern for PD patients. Reference 6 stated < 5% cited the presence of a catheter as being a factor against the decision to pursue PD which does not seem to support "many."

While the reviewer is correct that the studies do not specifically cite long catheter length as being the cause for self-image concerns (as opposed to the presence of any exposed catheter, regardless of length), the authors presume that minimizing the exposed catheter length would, at least in some way, alleviate these feelings. To make this statement clearer, and to qualify the causes of patient discomfort, this statement has been modified as follows:

“We identified one of the barriers (patient-related factor) to be the unnecessarily long PD transfer set protruding from their abdomen and affecting their self-image and daily life [6-8]. Between the catheter and transfer set, many PD patients have upwards of 16 cm of tubing protruding from their abdomen.”

“Peritoneal dialysis (PD) is an effective renal replacement strategy for patients with end-stage renal disease utilizing the peritoneum as the filter and PD catheter as access. A survey of PD patients showed that somemany felt uncomfortable with the length of current catheters and would be interested to explore newer, shorter designs.”

• As the height of the fluid column will change as dialysate flows from the dialysate bag to the patient's abdomen, should the calculations for hydrostatic pressure be reworked to reflect a dynamic fluid column rather than a static one?

The reviewer is absolutely correct that the varying fluid level within the dialysate bag will lead to varying hydrostatic pressures transmitted through the fluid column. The hydrostatic pressure of a fluid column is given by ΔP = ρgh. As the height of the dialysate (measured from bag bottom to bag top) is relatively small compared to the fluid column height (measured from patient’s abdomen to bag top), such a pressure change would also be relatively small. For example, a 20cm tall bag, elevated 1m above the patients abdomen would have a maximum pressure of (1m + 0.2m)ρg when full, and a minimum pressure of (1m)ρg, when empty. This results in a ~16% change in total pressure.

The manuscript has been clarified to illustrate this with the following addition:

“Inlet pressure was held constant as the changes in hydrostatic pressure from decreasing fluid level within the bag are small relative to overall pressure, and furthermore vary with bag dimensions.”

• Clarification needed for "typical real-world flow rate of approximately 0.2 L/min (lines 157-158); is this due to the 3mm cavity?

Thank you very much for pointing out the ambiguity of this section. The 0.2 L/min flowrate and the 3mm diameter together give the average fluid velocity, which is in turn used to calculate Reynold’s number. This description has been clarified as follows:

“Re = uD/ν

Reynolds number, denoted as Re, describes how turbulent a given flow is expected to behave. It is a function of fluid velocity u, hydraulic diameter D and kinematic viscosity ν. Fluid velocity is calculated from the volumetric flowrate and the flow’s cross-sectional area. Considering a typical real-world flow rate of approximately 0.2 L/min, the Reynolds number of such a flow through a 3 mm diameter cavity with a kinematic viscosity approximately equal to that of water at room temperature would be approximately 1400. As this Reynolds number well falls within the laminar regime, simulated flows were assumed to be laminar, and turbulent flow was not considered.

• The paper notes that the CFD was only on the helical prototype based on the CAD model for the helical design being used, but the conclusion as written appears to advocate for the accordion/expandable prototype, which was not subjected to a CFD.

Thank you very much for pointing out this inconsistency. The simulation has been re-run using the accordion model to remain consistent with the rest of the manuscript. All text and captions have been modified to reflect this.

---

## [Decision Letter · Decision Letter 1]

Addressing the Barriers to Peritoneal Dialysis - Visual Appeal Matters

PONE-D-24-35117R1

Dear Dr. Singh,

We’re pleased to inform you that your manuscript has been judged scientifically suitable for publication and will be formally accepted for publication once it meets all outstanding technical requirements.

Kind regards,

Ankur Shah

Academic Editor

PLOS ONE

Additional Editor Comments (optional):

Reviewers' comments:

Reviewer's Responses to Questions

**Comments to the Author**

Reviewer #2: All comments have been addressed

2. Is the manuscript technically sound, and do the data support the conclusions?

Reviewer #2: Yes

3. Has the statistical analysis been performed appropriately and rigorously?

Reviewer #2: Yes

4. Have the authors made all data underlying the findings in their manuscript fully available?

Reviewer #2: No

5. Is the manuscript presented in an intelligible fashion and written in standard English?

Reviewer #2: Yes

Reviewer #2: Thank you for addressing the initial set of concerns. It would benefit the paper to include the results of your survey. Otherwise I have no other questions.

**Do you want your identity to be public for this peer review?** For information about this choice, including consent withdrawal, please see our Privacy Policy

Reviewer #2: No

---

## [Editor Report · Acceptance letter]

PONE-D-24-35117R1

PLOS ONE

Dear Dr. Singh,

I'm pleased to inform you that your manuscript has been deemed suitable for publication in PLOS ONE. Congratulations! Your manuscript is now being handed over to our production team.

Kind regards,

on behalf of

Dr. Ankur Shah

Academic Editor

PLOS ONE